Manuscript prepared for Atmos. Chem. Phys.

with version 2014/05/15 6.81 Copernicus papers of the LaTeX class copernicus.cls.

Date: 5 December 2018

# Long-term simulation of the boundary layer flow over the double-ridge site during the Perdigão 2017 field campaign

**Johannes Wagner**[1], **Thomas Gerz**[1], **Norman Wildmann**[1], **and  Kira Gramitzky**[1]

[1]Deutsches Zentrum für Luft- und Raumfahrt, Institut für Physik der Atmosphäre, 82234 Oberpfaffenhofen, Germany

*Correspondence to:* Johannes Wagner (johannes.wagner@dlr.de)

**Abstract.** The Perdigão campaign 2017 was an international field campaign to measure the flow and its diurnal variation in the atmospheric boundary layer over complex terrain. A huge dataset of meteorological observations was collected over the double-hill site by means of state-of-the-art meteorological measurement techniques. A focus of the campaign was the interaction of the boundary layer flow with a single wind turbine, which was located on the south-western (SW) ridge top.

In this study a long-term nested large-eddy-simulation (LES) of 49 days duration with a maximum horizontal resolution of 200 m is used to describe both the general meteorological situation over Spain and Portugal and the local small-scale flow structures over the double-hill during the intensive observation period (IOP). The simulations show that frequently observed nocturnal low-level jets (LLJ) from NE have their origin over the slopes of the elevated plateau between the Portuguese Serra da Estrela and the Spanish Sierra de Gata mountain ranges N and NE of Perdigão and that the diurnal clockwise turning of the

wind direction over the double-ridge is induced by slope- and valley-winds under weak synoptic conditions. It is found that in spite of the long simulation time, modelled and observed wind structures on the ridge tops agree well, while along-valley flow within the valley is underestimated by the model.

## 1   Introduction

The generation of electrical power from wind turbines (WT) is a worldwide fast growing industry and a key technique to extend renewable energies (Emeis, 2013). Most of the areas that are available for onshore wind parks in flat terrain have already been exploited on the European continent and further wind farms need to be installed in topographically complex terrain (Schulz et al., 2014). The precondition to operate wind farms economically under these conditions is the ability to understand and simulate the planetary boundary layer (PBL) flow in complex terrain and its interaction with WTs (Tian et al., 2013). A large number of flow phenomena, such as thermally driven flows, gravity wave induced downslope wind storms and rotors are common over complex terrain and difficult to simulate with numerical models. Of special interest for wind park operators is the improved forecast of low-level jets (LLJs) (Storm et al., 2009). These are a worldwide phenomenon occuring where a local wind maximum is observed close to the ground and they can develop both over large areas (e.g., the Great Plains in the US, Rife et al., 2010) and very localized over complex terrain regions (e.g., within small valleys and basins, Banta et al., 2004). LLJs are important for the formation of heavy precipitation events and for the transport of dust and aerosols over large distances (Monaghan et al., 2010). Moreover, they are a significant source for wind power generation due to increasing WT hub heights. Due to the shallow structure of the mostly nocturnal jets, the correct simulation of LLJs with operational weather models is a challenge. It requires a sufficiently high horizontal and vertical grid resolution and a realistic representation of geographic features, such as topography, landuse and surface roughness in the models.

In order to provide a new data set of PBL-flow over complex terrain including the interaction with a single WT, the international field campaign Perdigão 2017 was organized in the context of the project "New European Wind Atlas" (NEWA, Mann et al., 2017) to measure the flow over a nearly parallel double-hill topography in the Portuguese back country (Fernando et al., 2018). The double-ridge site was chosen as it allows a smooth transition from idealized to complex terrain. This simplifies the application of both idealized and realistic numerical modelling. In addition, the region around Perdigão is known for its frequent occurence of diurnally changing NE and SW flow, which might be induced by thermally driven LLJs. The massive instrumentation during the intensive observation period (IOP) of the field campaign with i.a. up to 49 meteorological towers, 28 Doppler wind lidars and 6-hourly radiosonde launches, provided a huge data set of meteorological observations, which can be used to test and verify numerical simulations [1]. In this study a long-term large-eddy-simulation (LES) is used to characterize

---

[1]The experimental layout is described in detail at: perdigao.fe.up.pt

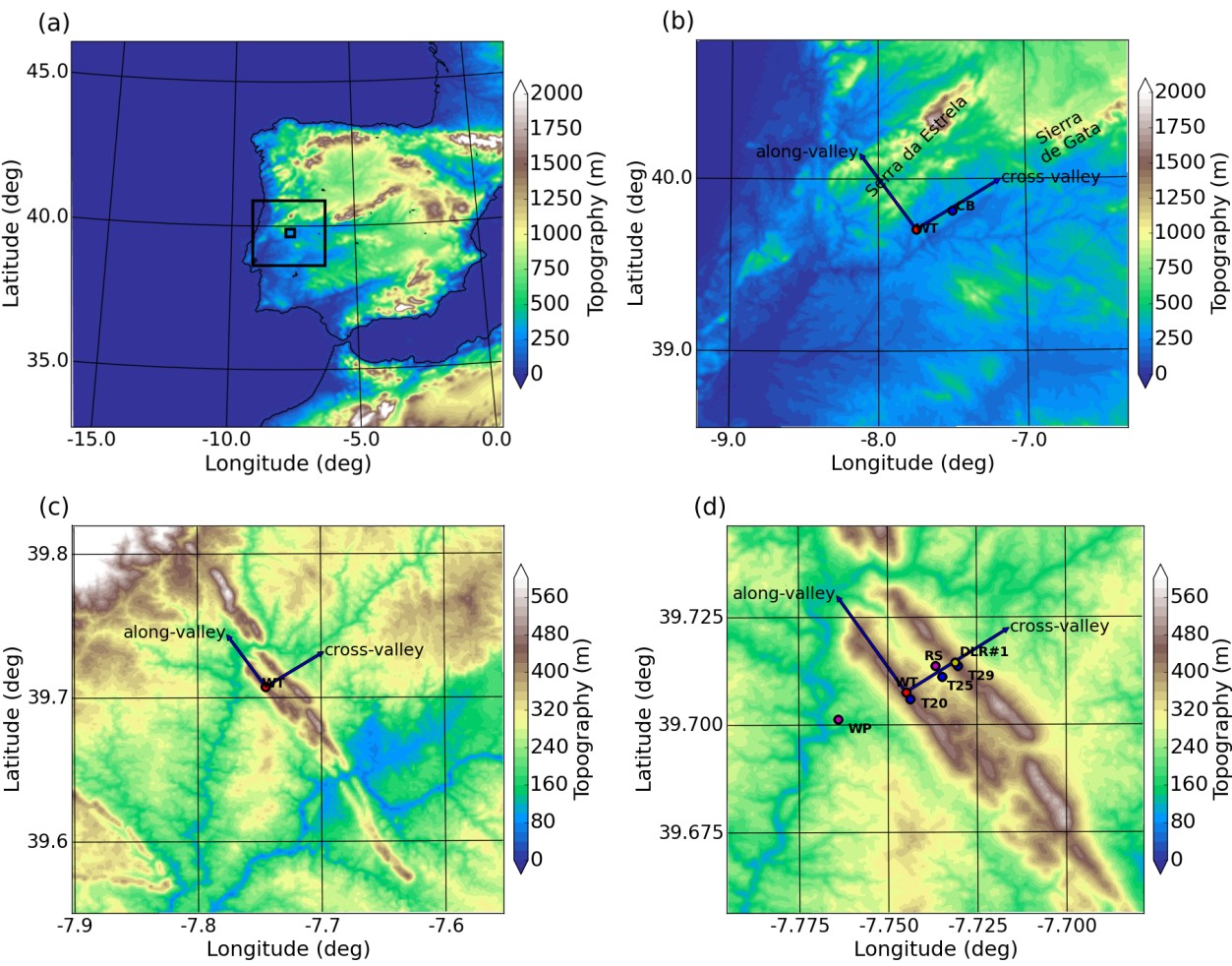

**Figure 1.** Topographic map of Spain and Portugal and operational area of the Perdigão field campaign. The shown areas in (a) mark the modelling domains D1 to D3. In (b) and (c) the topography of domain D2 and D3 is shown. The red dot marks the position of the wind turbine (WT) on the SW ridge and the town Castelo Branco (CB) is marked with a blue dot in (b). In (d) the double-ridge of domain D3 is enlarged to indicate the location of the WT, the three 100 m towers T20 (tower 20/tse04), T25 (tower 25/tse09) and T29 (tower 29/tse13), the Doppler wind lidar DLR#1, the wind profiler (WP) and the launch site of the radiosondes (RS). The blue perpendicular arrows in (b) to (d) mark cross- and along-valley wind directions. Cross-valley winds are defined by the location of the WT and the wind lidar DLR#1.

the meteorological conditions and to identify dominant flow patterns during the IOP with a focus on LLJ-events. Observational data are used to verify simulation results and to reveal potential for model improvement.

The paper is organized as follows: in section 2 the set-up of the numerical model is described. Dominating meteorological flow patterns during the field campaign are presented in section 3 and simulation results are compared to observations in section 4. Dominating LLJ-mechanisms are analysed in section 5 and a conclusion is given in section 6.

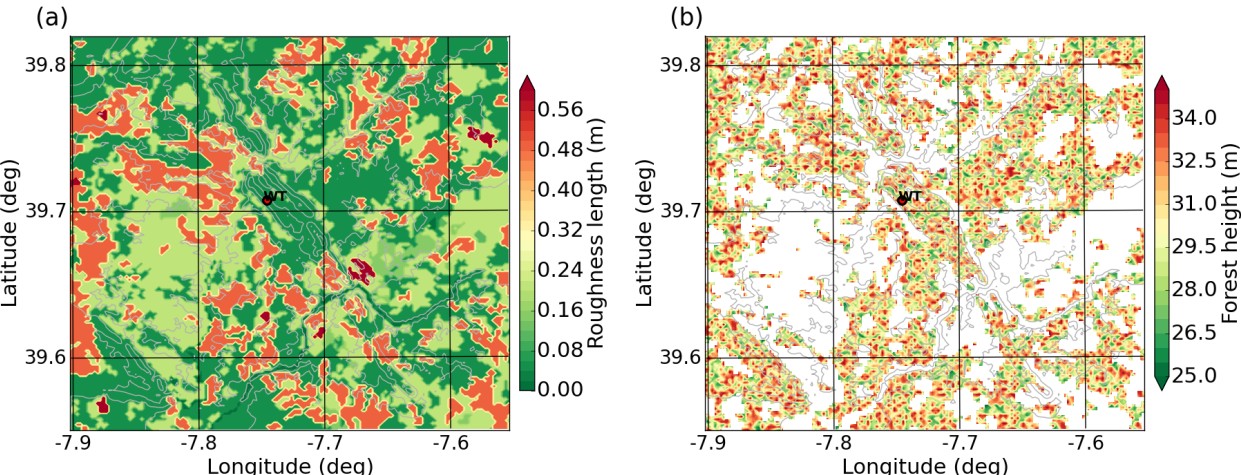

**Figure 2.** Surface roughness length of (a) the CORINE data set and (b) tree heights of the additionally implemented forest friction term applied to domain D3. White areas are not covered by forest. The location of the WT is marked with the red dot and the topography is indicated with grey contour lines.

## 2 Set-up of the numerical model

In this study, a long-term simulation is performed with the Weather Research and Forecasting (WRF) model version 3.8 (Skamarock et al., 2008). Three nested domains (D1, D2, and D3) with horizontal resolutions of 5 km, 1 km and 200 m are used (see Fig. 1). Domain D1 and D2 are run in RANS (Reynolds Averaged Navier Stokes) mode, while domain D3 is run in LES mode (see below for details on differences between these two modes). **A grid size of 200 m was necessary to resolve the double-ridge topography of Perdigão and its interaction with the PBL flow with at least 7 grid points (the distance between the two ridges is about 1.4 km).** The LES-setup was chosen to be independent of boundary layer parameterizations in domain D3 although a horizontal resolution of 200 m is relatively coarse for a LES run. Note that no mechanism was implemented in WRF to generate turbulence at the lateral edges of the LES domain, e.g., similar to the method described in Muñoz-Esparza et al. (2017). **The application of such methods requires considerably higher grid resolutions in the order of 50 m for convective and 10 m for stably stratified conditions (Muñoz-Esparza et al., 2017; Muñoz-Esparza and Kosović, 2018), which is not feasible for a long-term simulation. It is therefore not the focus of this study to simulate realistic turbulence over the double ridge, but to reproduce the mesoscale flow over complex terrain.** Vertical nesting is applied to define individual levels in the vertical for each model domain. This helps to avoid large grid aspect ratios near the surface and to save computational resources (Daniels et al., 2016). For domain D1 to D3 36, 57 and 70 vertically stretched

**Table 1.** Overview of the WRF model set-up. Variables indicate the horizontal resolution ($\Delta$x) and the minimum distance of levels in the vertical near the surface ($\Delta z_{min}$). The number of grid points in x, y, and z direction is marked with nx, ny and nz, respectively.

| Domain | $\Delta$x | $\Delta z_{min}$ | nx×ny×nz | PBL scheme | Topography | Landuse | Forest param. | Forest height |
|--------|-----------|------------------|-----------|------------|------------|---------|---------------|---------------|
| D1 | 5 km | 80 m | 300×300×36 | YSU | GTOPO30 | USGS | no | - |
| D2 | 1 km | 50 m | 251×251×57 | YSU | GTOPO30 | USGS | no | - |
| D3 | 200 m | 15 m | 151×151×70 | - | ASTER | CORINE | yes | 30 m±5 m |

levels are used and the respective lowest model levels are set to 80 m, 50 m and 15 m above ground level (AGL). The model top is defined at 200 hPa (about 12 km height) to include radiation and cloud effects at the tropopause. At the model top a 3 km thick Rayleigh damping layer (Klemp et al., 2008) is applied to prevent wave reflection. Physical parameterizations contain the Rapid Radiative Transfer Model longwave scheme (Mlawer et al., 1997), the Dudhia shortwave scheme (Dudhia, 1989), the Yonsei University (YSU) boundary layer scheme (Hong et al., 2006), the Noah land surface model (Chen and Dudhia, 2001), the WRF single-moment 5-class microphysics scheme (WSM5, Hong et al., 2004; Hong and Lim, 2006) and the Kain-Fritsch cumulus parameterization scheme (Kain and Fritsch, 1990). In domain D3 the boundary layer and cumulus schemes are switched off (LES mode) and subgrid-scale turbulence is parameterized by a three-dimensional 1.5 order turbulent kinetic energy (TKE) closure (Deardorff, 1980). The simulation is initialized once at 00 UTC 30 April 2017 and run for 49 days and 18 hours until 18 UTC 18 June 2017. The initial and boundary conditions are supplied by ECMWF operational analyses on 137 model levels with a horizontal resolution of 8 km and a temporal resolution of 6 hours. The WRF output interval of domain D3 was set to 10 minutes to allow a better comparison with tower measurements averaged over 10 minutes.

For domain D1 and D2, the Global 30 Arc-Second Elevation (GTOPO30) digital elevation model and the U.S. Geological Survey (USGS) landuse data set are used. These are provided by the WRF preprocessing system (WPS). For domain D3, the Advanced Spaceborne Thermal Emission and Reflection Radiometer (ASTER) topography data set (Schmugge et al., 2003) with a horizontal resolution of 30 m and the Coordination of Information on the Environment (CORINE) land cover data with a horizontal resolution of 100 m is used to better resolve the double-ridge topography and landcover of the Perdigão region. The CORINE landuse categories were transformed into the 24 USGS WRF landuse types according to Pineda et al. (2004). The inspection of surface roughness lengths from CORINE landuse data indicates that roughness lengths are considerably too small. For example, Fig. 2(a) shows that CORINE roughness lengths are in the order of 0.1 m over the double-ridge. In

reality the hills were partially covered by eucalyptus trees with heights of about 20 m to 25 m, which should be represented by roughness lengths in the order of 1 to 2 m. Short-term standard WRF simulations of LLJ-cases over Perdigão were run for

12 hours and showed that surface winds were clearly too high over the double-ridge region compared to lidar measurements (results will be shown in a successive paper). This was improved by the implementation of an additional friction term in the LES domain D3 in form of the forest parameterization described by Shaw and Schumann (1992), which acts on the lowermost model levels. The friction term was activated on grid points, which were classified as forest in the CORINE landuse data set (see Fig.2(b)). Tree heights in these forest areas were randomly distributed by 30 m $\pm$ 5 m and were defined somewhat higher

in the model compared to real tree heights (about 25 m) to ensure that at least the lowermost 2 to 3 model levels are located within the canopy layer. An overview of the described model set-up is shown in Table 1.

## 3   Meteorological flow patterns during the field campaign

The IOP of the Perdigão field campaign took place from 1 May to 15 June 2017 (Fernando et al., 2018). The instrumentation was based on 49 meteorological towers (UCAR/NCAR - Earth Observing Laboratory, 2017a) with heights between 10 m to

100 m, more than 180 sonic anemometers, 21 scanning and 7 profiling wind lidars (e.g., Wildmann et al., 2018b), 3 microwave radiometers (MWR), a radio acoustic sounding system (RASS) wind profiler (WP, UCAR/NCAR - Earth Observing Laboratory, 2017b) and 6-hourly radiosonde (RS, UCAR/NCAR - Earth Observing Laboratory, 2018) launches. On the southwestern (SW) ridge, an Enercon E-82 2 MW WT with a hub-height of 78 m and a rotor diameter of 82 m is located (see Fig. 1). Sound propagation and immission was measured by 9 microphones on the up- and downstream side of the SW ridge. In this

study, data of the wind profiler (WP), radiosonde (RS) observations and tower measurements of tower T20 (tower 20/tse04), T25 (tower 25/tse09) and T29 (tower 29/tse13) were used (see Fig. 1). As the main wind directions at Perdigão are NE and SW-flow, the focus in this study is on flow perpendicular (cross-valley) and parallel (along-valley) to the double-ridge. The direction of cross- and along-valley winds is marked with the blue arrows in Fig. 1(b) to (d). Cross-valley winds were defined along the cross-section of the DLR#1 wind lidar and is therefore not perfectly perpendicular to the double-ridge. Along-valley

winds are defined to be perpendicular to cross-valley winds. Negative cross- and along-valley winds mean winds from NE and NW directions, respectively.

To give an overview of the meteorological conditions during the campaign, a WRF meteogram of domain D3 for the location of the WT is shown in Fig. 3. Based on the synoptic conditions, the campaign can be divided into two phases. During the first

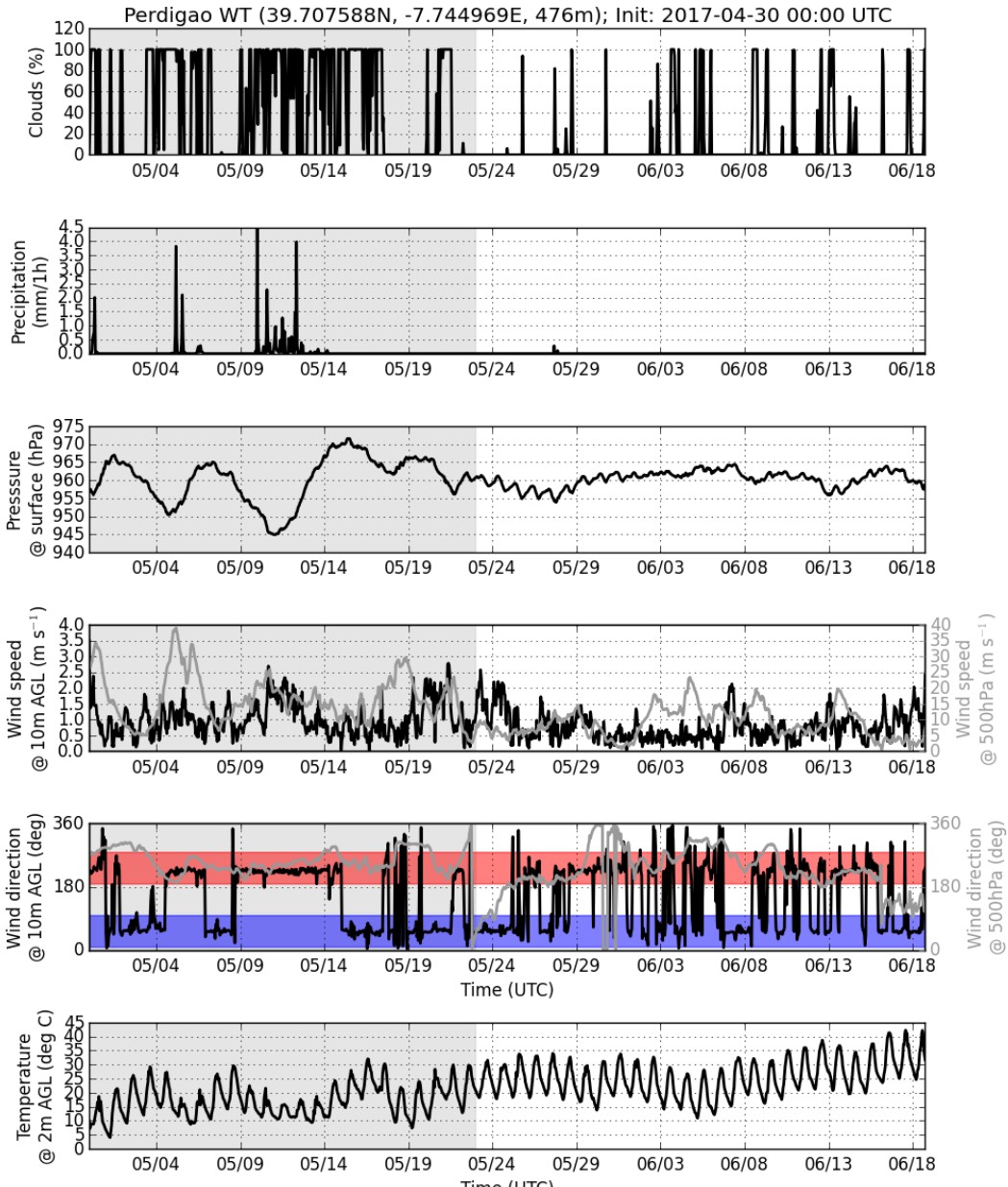

**Figure 3.** WRF D3 meteogram for 30 April to 18 June 2017 at the location of the WT on the SW ridge. The red and blue shaded areas in the wind direction plot mark cross-valley surface winds from SW and NE directions, respectively. The grey shading separates the campaign synoptically into phase I and phase II.

phase from 1 May to about 23 May the Perdigão region was influenced by periodic passages of low and high pressure systems, as is visible from the surface pressure time series. Low pressure systems were accompanied by precipitation events, increased cloudiness, reduced diurnal surface temperature variation and cross-valley winds at 10 m AGL from SW (wind direction sector

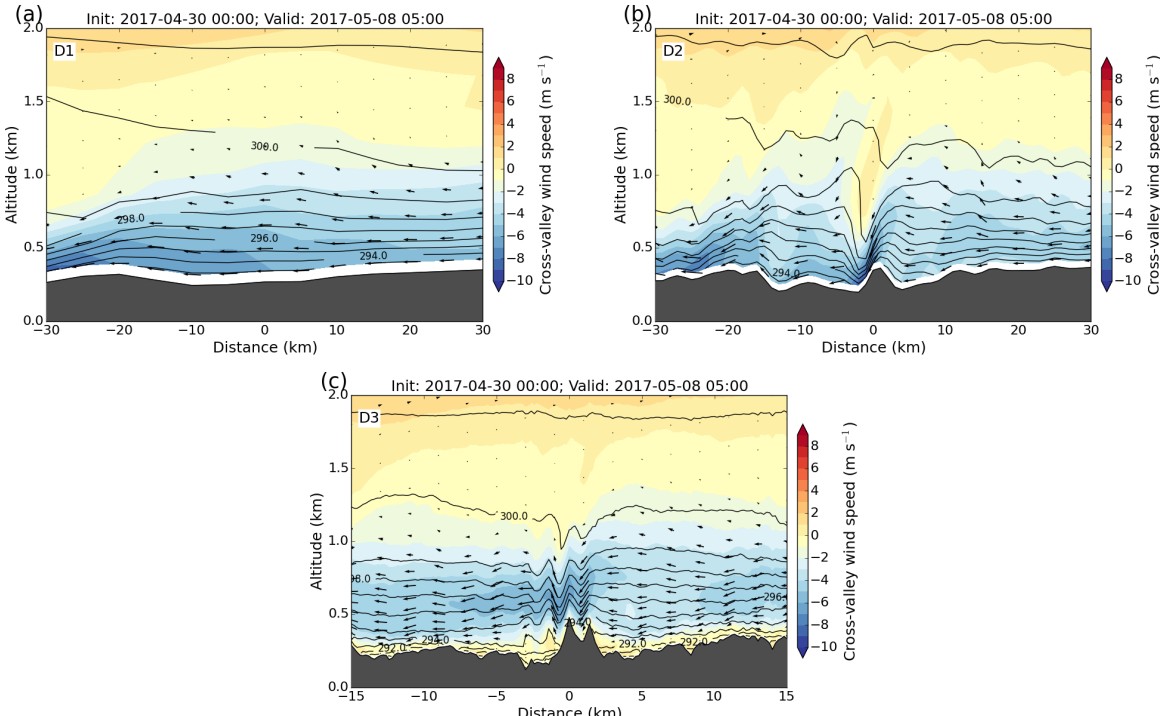

**Figure 4.** Vertical cross-sections of cross-valley wind speed for WRF domains D1 to D3 valid at 05:00 UTC 8 May 2017. The cross-section is centred at the location of the WT and is oriented in cross-valley direction (see Fig. 1). In (c) only the half horizontal distance is shown compared to (a) and (b).

marked with red shading in Fig. 3). During high pressure events, wind speed and wind direction at 500 hPa represent the conditions in the free atmosphere and show that winds were blowing from western directions throughout the first phase of the campaign. Surface winds were decoupled from the free atmosphere during the night and thermally driven LLJs from NE developed (wind direction sector marked with blue shading in Fig. 3), e.g., in the period from 2 to 3 May, 7 to 8 May or 16 to 17 May. As an example, a typical LLJ-case on 8 May 2017 at 05 UTC is shown in Fig. 4 by means of vertical cross-sections of cross-valley wind for WRF domains D1 to D3. Similar situations occured frequently during the campaign. In all three domains strong winds from NE are visible, however, only in domain D3 the double-ridge is resolved and a LLJ including lee-waves is generated. In D1 and D2 the topography is strongly smoothed and no LLJ develops due to strongly overestimated winds near the surface. In D1 the Perdigão topography is nearly flat and in D2 only one single hill exists. In both domains no valley boundary layer including recirculation and along-valley flows can develop and results in considerable deviations from observed winds (see following section).

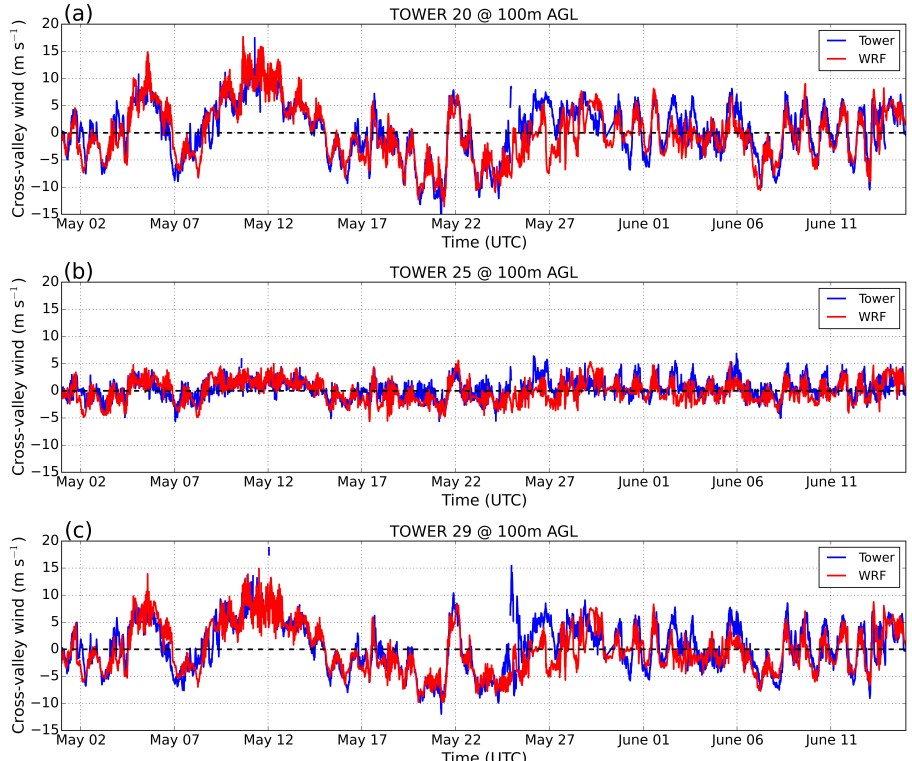

**Figure 5.** Time series of observed and simulated (WRF D3) cross-valley wind at 100 m AGL for (a) to (c) tower T20, T25 and T29 (see Fig. 1 (c) for tower locations).

During the second phase starting at about 23 May (see Fig. 3), the subtropical tropospheric jetstream has moved further north and the Iberian Peninsula was located under stable high pressure conditions. During this phase, cloud coverage and rain events decreased significantly while diurnal surface temperature variations and daily temperature maxima increased until the end of the campaign. In the free atmosphere, winds were weaker in comparison to the first phase and there were cases with non-western wind directions at 500 hPa (e.g., 23 May, 28 May, 12 June). Surface winds were often supergeostrophic compared to winds at the 500 hPa level and showed a more frequent diurnal variation of SW and NE winds compared to the first phase. This can be explained by the development of thermally driven wind systems, which were favoured under weak synoptic conditions and are analysed in more detail in section 5.

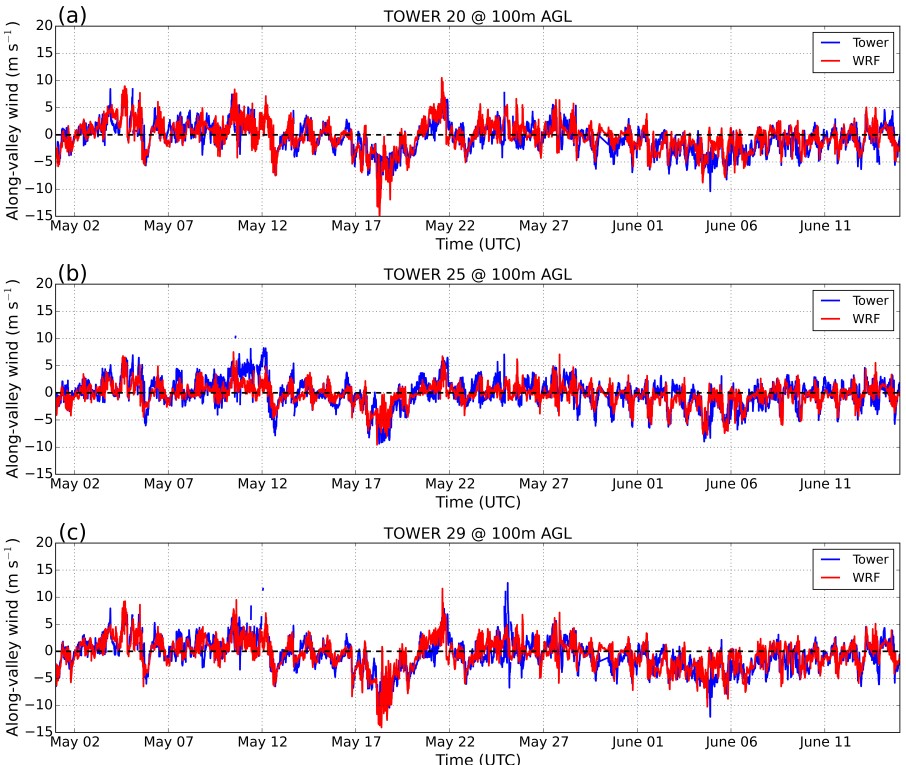

**Figure 6.** As in Fig. 5, but for the along-valley wind component.

## 4   Model verification

The dataset of remote and in-situ observations obtained during the Perdigão campaign enables to evaluate the long-term simu-

lation. Figure 5 and 6 show observed and simulated WRF D3 time series of cross- and along-valley wind at 100 m AGL at the

locations of tower T20, T25 and T29 (see Fig. 1 for the tower locations). The output interval of both data sets was 10 minutes.

Simulated cross- and along-valley winds show very good agreement with all three tower stations. Both absolute values and

phase of the observed signal are reproduced well by the model. It has to be recapitulated that the model was only initialized

once and was run for a period of 49 days. The lateral boundaries of domain D1, where ECMWF data serve as boundary condi-

tions, are 750 km away from the Perdigão site (see Fig. 1 (a)). Within the WRF domains the model develops its own dynamics

and is capable to reproduce the diurnally changing flow systems during the whole simulation period with a surprisingly high

quality. The comparison of tower T20 and T29 in Fig. 5(a) and (c) shows similar time series of cross-valley winds, as these

towers were located on the SW and NE ridge, respectively and probed the PBL at the same altitude. Especially during phase II,

diurnally changing NE and SW flow is visible from both T20 and T29 measurements and model simulations. Due to the loca-

**Table 2.** Correlation coefficients (COR) and root-mean-square-error values (RMSE) for the comparison of WRF data with observations at tower T20, T25 and T29. Values for cross- (along-) valley winds are written in bold (normal) font.

| | T20 | | T25 | | T29 | |
|---|---|---|---|---|---|---|
| Domain | COR | RMSE (m s$^{-1}$) | COR | RMSE (m s$^{-1}$) | COR | RMSE (m s$^{-1}$) |
| D1 | **0.802**; 0.756 | **3.08**; 2.48 | **0.552**; 0.675 | **4.06**; 3.02 | **0.775**; 0.748 | **3.18**; 2.54 |
| D2 | **0.817**; 0.748 | **3.06**; 2.31 | **0.582**; 0.739 | **4.04**; 2.17 | **0.785**; 0.752 | **3.21**; 2.29 |
| D3 | **0.822**; 0.744 | **2.95**; 1.90 | **0.612**; 0.694 | **1.74**; 1.94 | **0.776**; 0.746 | **2.96**; 1.97 |

tion within the valley, Tower T25 in Fig. 5(b) shows much weaker cross-valley winds. Along-valley winds (Fig. 6) at T20 and T29 on the ridge tops are significantly lower than cross-valley winds on the ridge tops. In the valley at T25 along-valley winds seem to be the dominant wind component and reveal a diurnal changing NW and SE flow. For both cross- and along-valley winds correlation coefficients and root-mean-square-errors (RMSE) were computed for T20, T25 and T29 for WRF domains D1 to D3 (see Table 2). The highest correlations are found on the ridge tops (T20 and T29) and surprisingly the correlation coefficients are very similar for all WRF domains. This means that the phase of the (diurnally) changing cross- and along-valley wind signals can be reproduced by the two coarse model domains D1 and D2 although they do not resolve the double-ridge. This can be explained by the fact that changing NE- and SW-winds are thermally driven by a mesoscale horizontal pressure gradient (see next section), which is resolved by D1 and D2. The high grid resolution of domain D3 improves local effects over the double-ridge, but does not increase the correlation. The correlation coefficient alone is, however, not sufficient to assess the model performance. Looking at RMSE-values domain D3 reveals considerably reduced errors (up 2.3 m s$^{-1}$ at T25) compared to the coarser resolved domains. This is due to the better representation of topographically induced flow patterns, especially within the valley (T25).

To analyse the distribution of horizontal wind speed and direction at the three towers, corresponding wind roses are plotted in Fig. 7. The dominant wind directions from measurements at T20 and T29 on the two ridge tops were NE and SW (Fig. 7 (a) and (c)). For these two sites the WRF D3 distributions are in good agreement with observations (Fig. 7 (d) and (f)). For T25 at the valley floor, WRF D3 favours wind directions from NE, while observed directions show peaks for NW and SSE wind directions. This means that wind directions at T25 are not well represented by domain D3 while the magnitude of wind speeds at the valley floor are comparable to observed values. The reason for underestimated along-valley flow is not clear and further

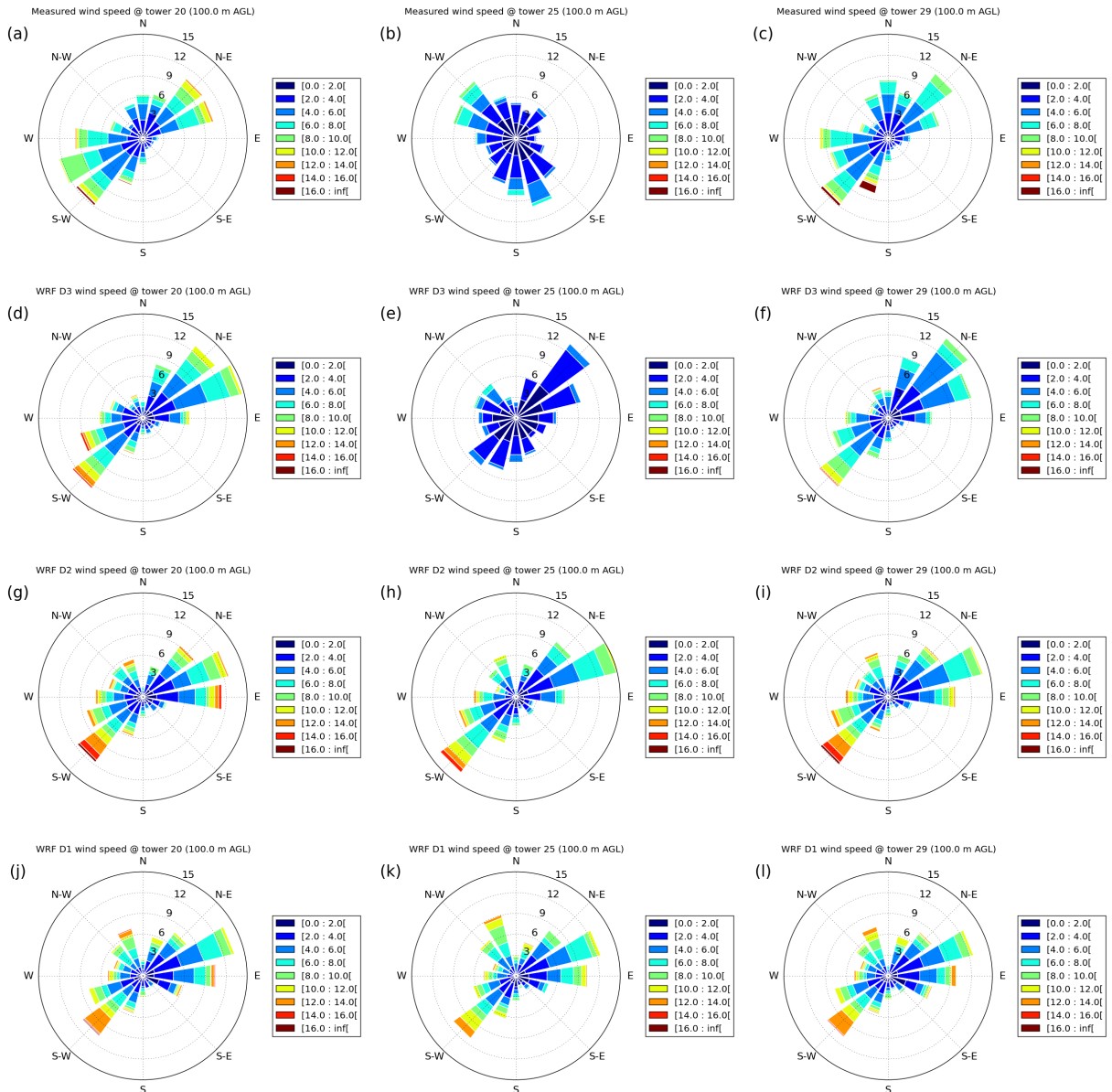

**Figure 7.** Wind distribution at 100 m AGL for observed (a), to (c) and simulated horizontal winds for (d) to (f) domain D3, (g) to (i) domain D2 and (j) to (l) domain D1. The columns from the left to the right correspond to the locations of tower T20, T25 and T29, respectively (see Fig. 1 (c) for tower locations).

sensitivity runs are necessary to investigate this issue. Windroses for domain D2 and D1 are plotted in Fig. 7(g) to (l). For all three tower locations the wind distribution is very similar for the respective model domain D1 and D2 due to the missing double-ridge topography. The dominance of the SW and NE wind directions at T20 and T29, which is visible in observed and D3 wind roses is less pronounced in D2 and D1 simulations, as other wind directions frequently occur in the range from

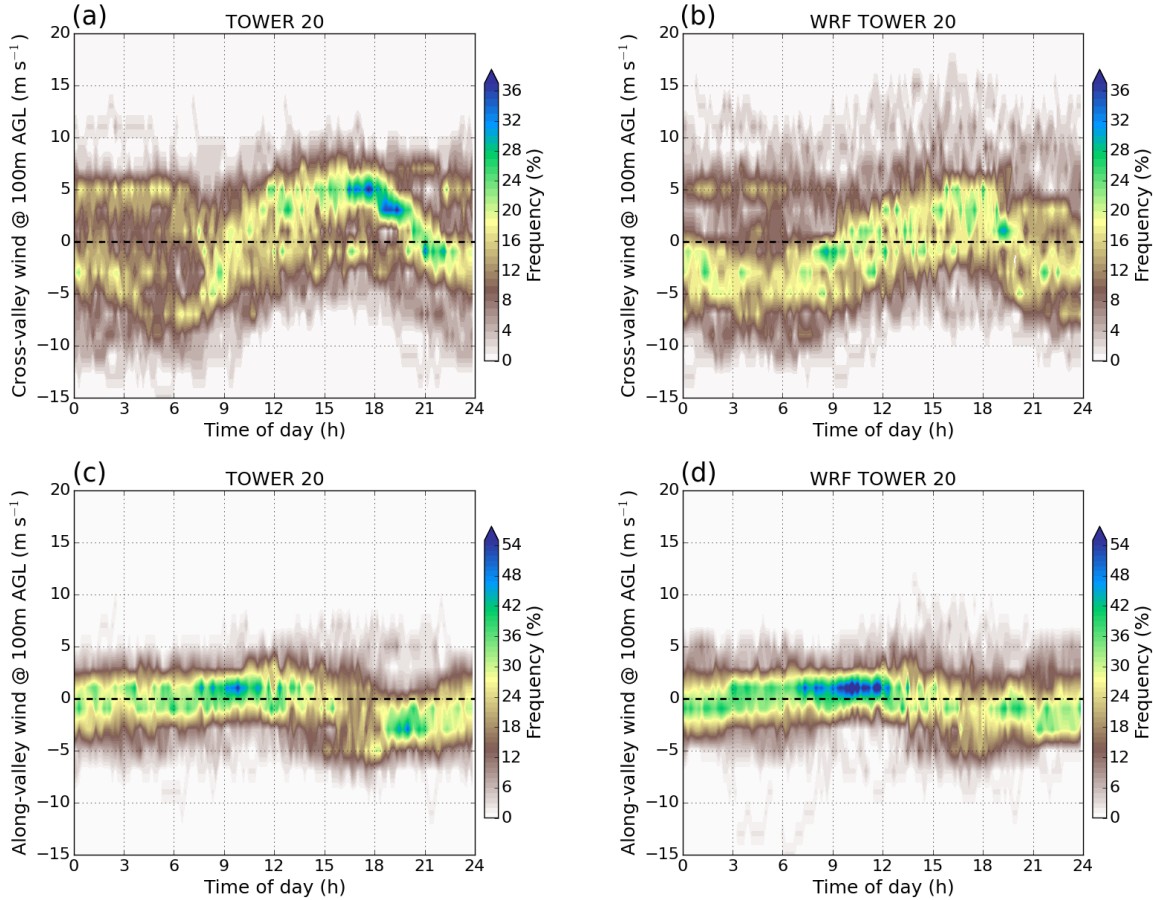

**Figure 8.** Diurnal distribution of (a) and (b) cross- and (c) and (d) along-valley wind at 100 m AGL at tower T20 in the period 1 May to 15 June 2017. Shown are observed tower data in (a) and (c) and data from WRF D3 simulations in (b) and (d).

WNW to NNW. Wind speeds are generally too strong in D1 and D2 due to overestimated surface winds. This is also shown by large RMSE values in Table 2 and exemplarily by means of the LLJ-cross-sections in Fig. 4. Largest differences between observations and D1 and D2 wind distributions occur at the valley station T25.

As the simulation period was characterized by calm synoptic conditions (especially during phase II) and thermally driven flows played an important role (see next section), the daily distribution of cross- and along-valley winds at T20 and T25 is plotted in Fig. 8 and Fig. 9. The most distinct feature in Fig. 8(a) and (b) is the sinusoidal cross-valley wind distribution at T20. This is induced by nocturnal LLJs from NE and flow from SW to WSW at daytime. Cases with SW-flow during the night or NE-flow during the day were synoptically driven events. Along-valley winds at T20 show less variation as compared to cross-valley winds, but there is also a phase-shifted sinusoidal diurnal variation visible. Due to the influence of the Estrela mountains, minimum along-valley winds (NW-flow) typically occured during the late afternoon as will be described in the next

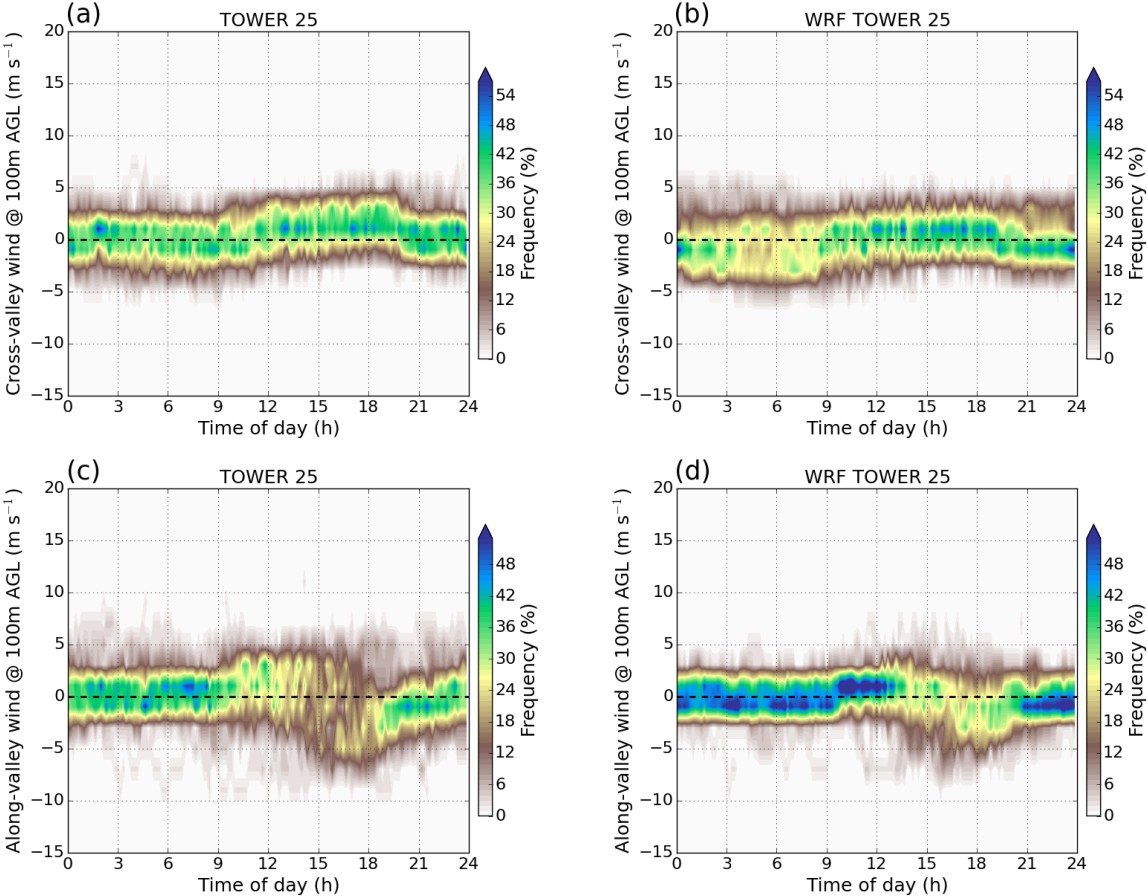

**Figure 9.** As in Fig. 8, but for tower T25 at the valley floor.

section. This diurnal variation in along-valley winds is more pronounced at T25 in the valley, as can be seen in Fig. 9(c) and

(d). At this site, the along-valley flow is the dominant flow feature. The comparison of observed and simulated along-valley

winds in Fig. 9(c) and (d) shows that WRF D3 computes the measured diurnal cycle, but along-valley winds are generally too

weak. This is in agreement with the wind roses in Fig. 7(b) and (e).

## 5    Low-level jet analysis

To analyse the occurence of LLJs during the campaign in more detail, the LLJ-index definition of Rife et al. (2010) for nocturnal

jets was used. This index is a measure for the strength of LLJs and is defined as:

$$\text{NLLJ} = \lambda\varphi\sqrt{[(u_{00}^{L_1} - u_{00}^{L_2}) - (u_{12}^{L_1} - u_{12}^{L_2})]^2 + [(v_{00}^{L_1} - v_{00}^{L_2}) - (v_{12}^{L_1} - v_{12}^{L_2})]^2},\tag{1}$$

with zonal and meridional wind components u and v at vertical levels AGL $L_1$ and $L_2$ at local times (LT) 00 LT and 12 LT.

The binary masks $\lambda$ and $\varphi$ ensure nocturnal and jet-like wind profiles and are defined as:

$$\lambda = \begin{cases} 0, & \mathrm{ws}_{00}^{L_1} \leq \mathrm{ws}_{12}^{L_1} \\ 1, & \mathrm{ws}_{00}^{L_1} > \mathrm{ws}_{12}^{L_1}, \end{cases} \tag{2}$$

and

$$\varphi = \begin{cases} 0, & \mathrm{ws}_{00}^{L_1} \leq \mathrm{ws}_{00}^{L_2} \\ 1, & \mathrm{ws}_{00}^{L_1} > \mathrm{ws}_{00}^{L_2}, \end{cases} \tag{3}$$

with horizontal wind speed ws at level $L_1$ and $L_2$, respectively. As we were not only interested in night-time jets, we neglect

the coefficient $\lambda$ and applied the following modified version of the LLJ-index based on hourly WRF data:

$$\mathrm{LLJ} = \varphi \sqrt{(\mathrm{u}^{L_1} - \mathrm{u}^{L_2})^2 + (\mathrm{v}^{L_1} - \mathrm{v}^{L_2})^2}, \tag{4}$$

with vertical levels $L_1$=300 m AGL and $L_2$=2000 m AGL. $\varphi$ is defined as:

$$\varphi = \begin{cases} 0, & \mathrm{ws}^{L_1} \leq \mathrm{ws}^{L_2} \\ 1, & \mathrm{ws}^{L_1} > \mathrm{ws}^{L_2}, \end{cases} \tag{5}$$

In contrast to Rife et al. (2010) our LLJ-index is just based on wind speed differences between level $L_1$ and $L_2$ and not on

wind speed differences between night and day. This enables to compute the index for each hour of the day and compare LLJ-indices during day- and night-time. Because of measured evidence and due to a higher horizontal and vertical grid resolution, shallow LLJs are well represented in our model set-up and we therefore defined the levels $L_1$ and $L_2$ at lower altitudes than Rife et al. (2010), who used $L_1$=500 m AGL and $L_2$=4000 m AGL with a horizontal model grid resolution of 4 km on 28 vertical levels. Figure 10(a) and (b) show temporal averages of the LLJ-index for the period 01 May to 15 June 2017 for night-

and day-time. The computation of night-time LLJ-indices was based on averaging times between 21 UTC to 9 UTC, while day-times were averaged between 9 UTC to 21 UTC. This classification was used as nocturnal LLJs were often observed to persist until after sunrise, while thermally driven winds from SW were observed until about 21 UTC. Note that local time in Portugal is UTC+1h during the summer season. The spatial distribution of nocturnal LLJs in Fig. 10(a) shows that the strongest nocturnal LLJs develop SW of Perdigão with a strong northerly flow down the slopes of the Serra da Estrela mountains towards

the flat Tejo-basin. In addition, strong jet winds developed NE of Perdigão over the slopes between the Serra da Estrela and

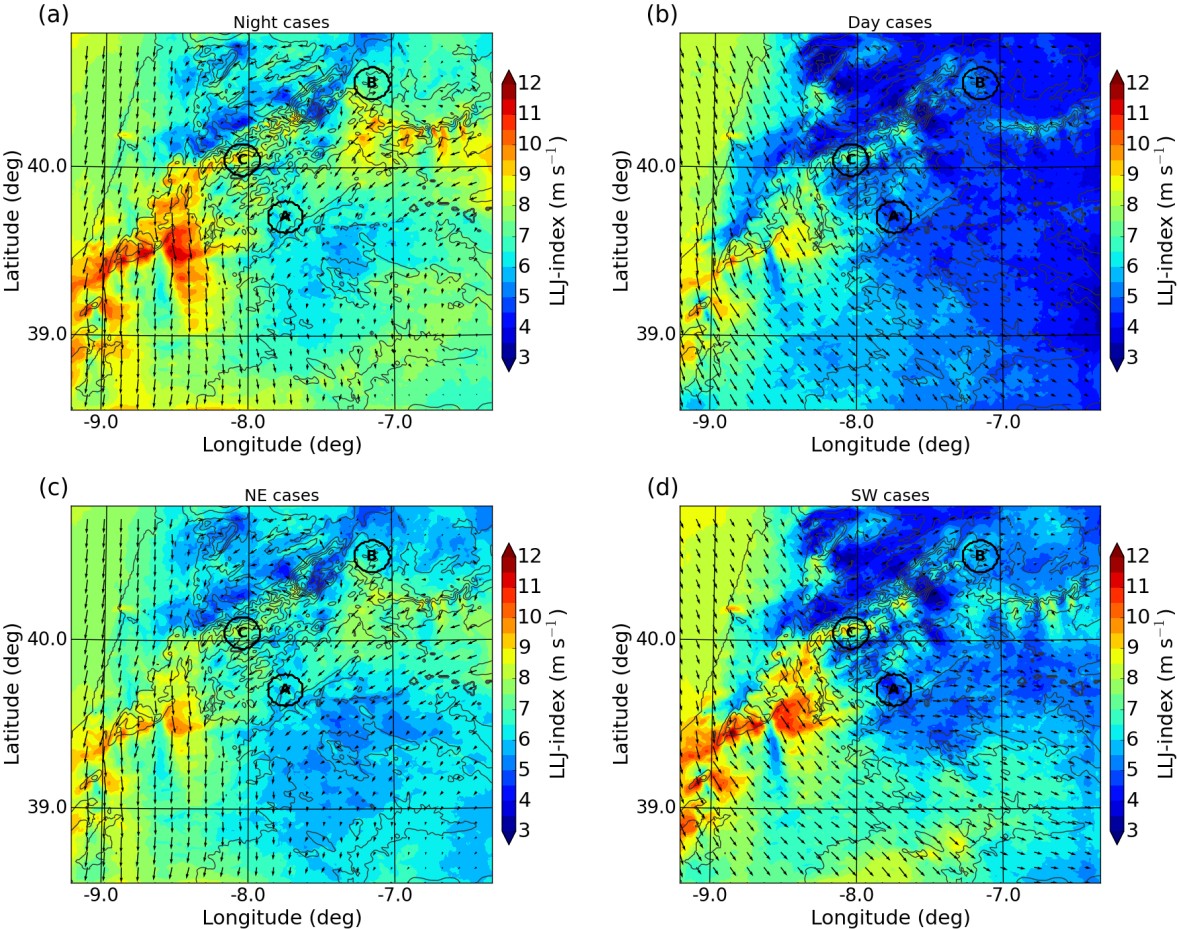

**Figure 10.** Temporal average of LLJ-index for the period 01 May to 15 June 2017. LLJs are detected by comparing wind speeds at 300 m AGL and 2000 m AGL during (a) night times (21 UTC to 9 UTC) and (b) day times (9 UTC to 21 UTC). In (c) and (d) only cases with wind directions from NE (negative cross-valley winds) and SW (positive cross-valley winds) at 300 m AGL at the location of the WT were used. Arrows show the mean wind field at 300 m AGL during (a) night-time, (b) day-time, (c) NE and (d) SW LLJ cases. Wind vectors are plotted on every 10th grid point. Areas marked by black circles and letters A to C indicate regions used for the computation of pressure and temperature gradients in cross- and along-valley direction (see text). Region A is centred at the location of the WT. The topography is indicated with thin black contour lines. Data is based on WRF domain D2.

the Sierra de Gata mountain ranges and flowed down the basin of Castelo Branco (see Fig. 1(b)) as a drainage flow towards the double-ridge of Perdigão. Over the sea there is also a significant LLJ-index signal with mean winds from the North. This flow is associated with the Azores anticyclone, which induces northerly winds parallel to the Portuguese coast line. During the day the LLJ-activity is strongly reduced NE of Perdigão as can be seen in Fig. 10(b). SW of Perdigão and over the ocean, the LLJ-index remains strong as during the night. In these regions, winds are blowing more from NW-directions during the day

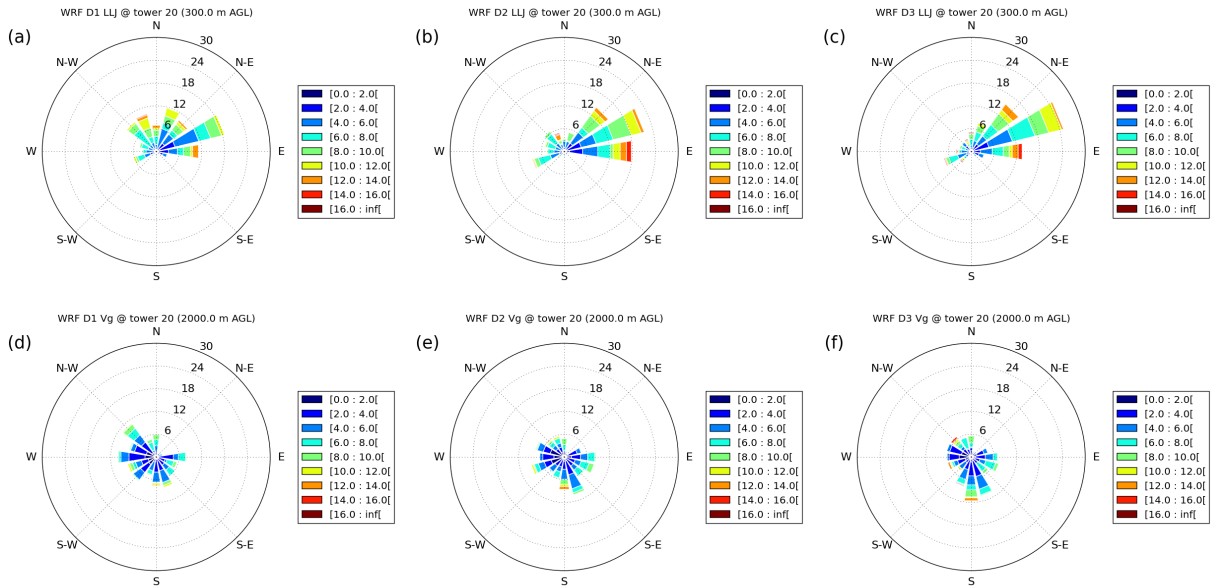

**Figure 11.** Distribution of winds during LLJ-events at (a) to (c) $L_1$ (300 m AGL) and at (d) to (f) $L_2$ (2000 m AGL) at the location of tower T20 for domains D1 to D3.

compared to northerly winds during the night. This is probably due to the sea breeze effect, which impresses an onshore wind component to the flow.

As the dominant wind directions for Perdigão are NE and SW winds, we additionally computed the mean LLJ-index for cases when the cross-valley wind at the location of tower T20 next to the WT was negative at 300 m AGL (NE cases) and

215 for cases when it was positive (SW cases). The corresponding LLJ-index maps are shown in Fig. 10(c) and (d). The spatial distribution of LLJs and the mean flow pattern during NE and SW cases is nearly identical to that during night and day cases, respectively (Fig. 10(a) and (b)). This is confirmed by a correlation coefficient of 0.92 for the pointwise comparison of night and NE LLJ-indices and by a correlation coefficient of 0.90 for day and SW cases. This indicates that on average, LLJs from NE and SW over the double-ridge were respective night- and day-time phenomena although, exceptional cases with nighttime

LLJs from SW have also been observed (Wildmann et al., 2018a). The spatial distribution of LLJ-winds at $L_1$ (300 m AGL) is plotted for the location T20 in Fig. 11(a) to (c) for D1 to D3. Nearly all LLJ-cases exhibit NE wind directions and have their origin over the slopes between the Serra da Estrela and Sierra de Gata mountain ranges (see Fig. 10(a), (c)). LLJs from SW are rare events. The LLJ-distribution for D2 is very similar to D3. For D1 LLJs are considerably weaker and there is a larger spread of wind directions. In Fig. 11(d) to (f) the distribution of winds during LLJ-events at $L_2$ (2000 m AGL) is shown at the

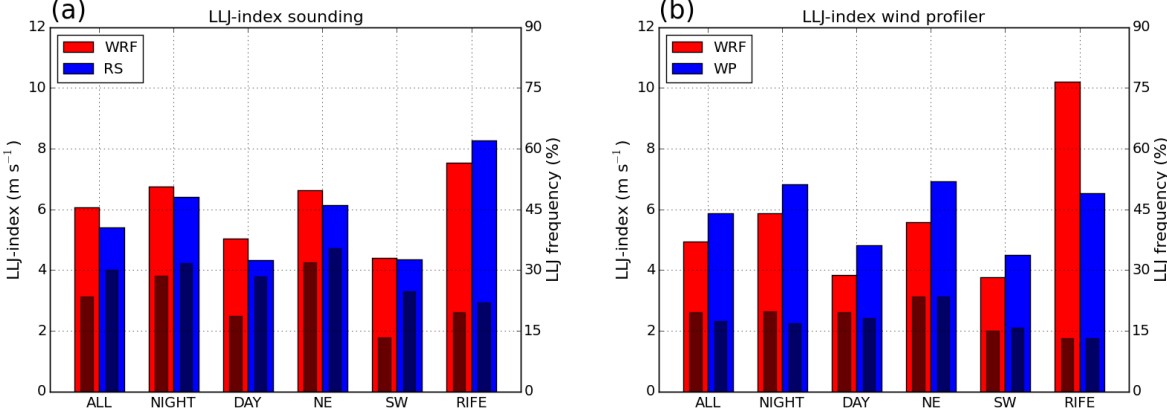

**Figure 12.** LLJ-indices obtained from (a) radiosonde (RS) launches and (b) wind profiler (WP) measurements during the campaign. WRF D3 indices were obtained by interpolating data in space and time on the radiosonde flight track and on the wind profiler location. Indices were averaged over the whole campaign period (ALL), the night and daytime (NIGHT, DAY) and for cases with wind directions from NE (negative) and SW (positive cross-valley winds at 300 m AGL). The nocturnal index according to the computation of Rife et al. (2010) is labeled with RIFE. The black bars indicate the frequency of LLJ occurence during all, night/day and NE/SW cases, respectively. For RIFE the black bars indicate the occurence of nocturnal jets during the whole campaign. See Fig. 1 for the wind profiler location and the launch site for soundings (every 6 h).

location of T20. These winds can be regarded as geostrophic background winds in the free atmosphere and indicate very calm conditions during LLJ events. Winds are quite regularly distributed between directions from SE to NW, which means that they were decoupled from the boundary layer and cannot be the main driving mechanism for the LLJs near the surface.

LLJ-indices were also computed for all radiosonde (RS) and wind profiler (WP) measurements during the campaign. WRF D3 data were interpolated in space and time on the RS flight track and on the WP location before the LLJ-index was calculated according to Eq. 4. Fig. 12 shows the direct comparison of simulated and observed mean LLJ-indices, which were averaged for all, night/day and NE/SW cases, respectively. Both RS and WP data indicate that LLJs were strongest during nighttime and during NE cases. The same result is obtained from WRF D3 simulations whose indices agree well with RS and WP observations. LLJ-events occured during 30.0% (WRF: 23.5%) of all RS- and during 17.4% (WRF: 21.9%) of all WP-observations during the campaign. Winds from NE at 300 m AGL (L1) were observed during 49.5% (WRF: 55.0%) of RS- and during 42.0% (WRF: 55.1%) of WP-measurements. The observed LLJ frequency for NE cases was higher for RS observations (35%, WRF: 32%) compared to WP-data (23%, WRF: 26%), as RS were launched within the valley (see Fig. 1)

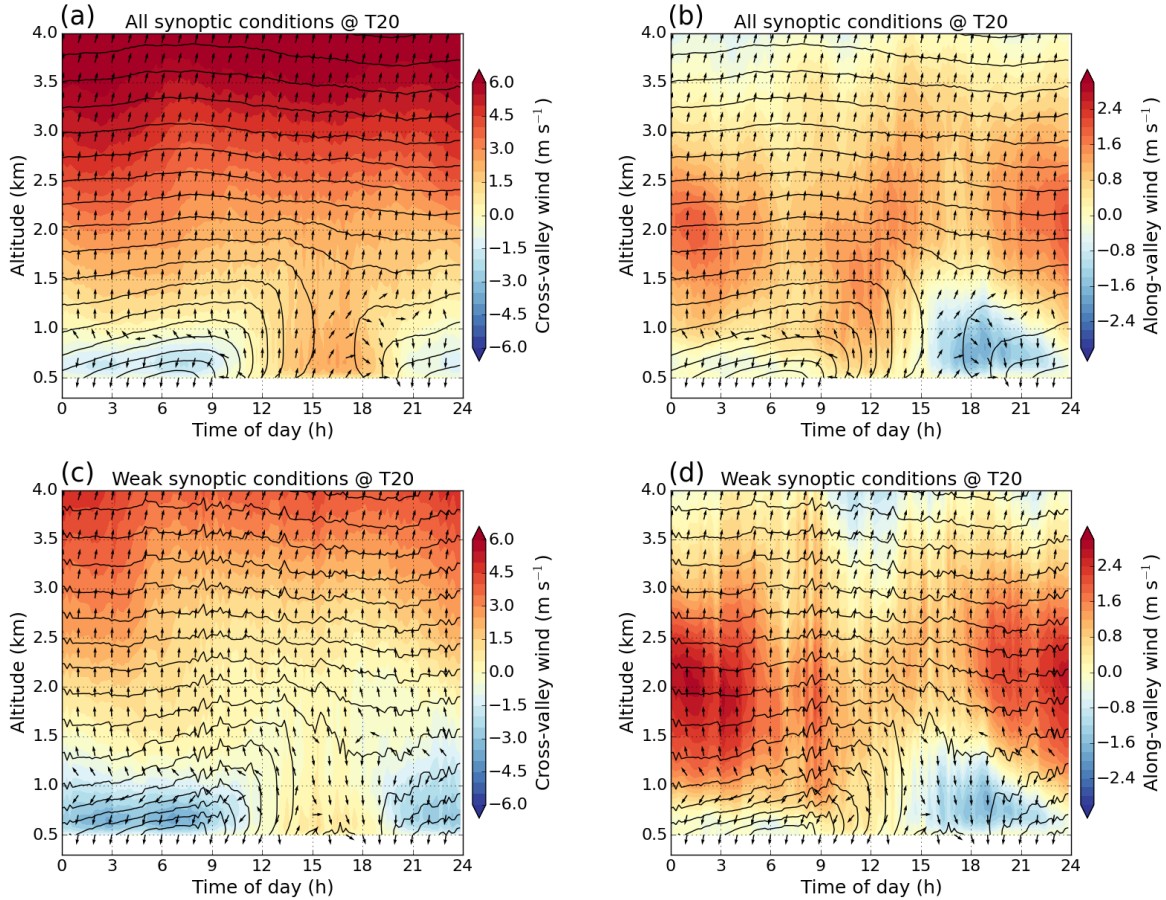

**Figure 13.** WRF D3 mean diurnal vertical profiles of (a) and (c) cross- and (b) and (d) along-valley wind at tower T20. The averaging period was 1 May to 15 June 2017. In (a) and (b) all data of this period were used for averaging, while in (c) and (d) only cases with weak synoptic conditions (defined by horizontal wind speeds < 10 m s$^{-1}$ at 3000 m altitude) were utilized. Negative cross- (along-) valley winds indicate flow from NE (NW) directions, respectively. Black arrows show the mean horizontal wind direction (North=top). Black contour lines mark isentropes with a contour interval of 1 K.

while the WP was located on the lee-side of the double-ridge. At tower T20, which is located on the SW ridge, LLJs did occur more frequently due to the elevated position during 42% of NE cases and during 30% of all synoptic conditions in the WRF D3 simulation (not shown). The original nocturnal NLLJ-index of Rife et al. (2010) (Eq. 1) was also computed and is labeled with RIFE in Fig. 12. NLLJ-values are slightly higher compared to our mean LLJ-indices probably as only one value per day is computed for NLLJ-indices by comparing the situation at 00 UTC and 12 UTC (see Eqs. 1 to 3). This results in a lower number of LLJ-events and a reduced smoothing when averaging is done over the whole campaign period. This can maybe explain the extreme high WRF NLLJ-index at the WP location.

To illustrate the daily changing flow patterns over the double-ridge in more detail, we plotted the simulated mean diurnal cycle of cross- and along-valley wind vertical profiles at the location of tower T20 in Fig. 13. In (a) and (b), averaging was done over the whole IOP from 01 May to 15 June 2017 and in (c) and (d) only for weak synoptic conditions. The latter were defined if the horizontal wind speed at an altitude of 3000 m was smaller than 10 m s$^{-1}$ to reduce the effect of synoptic forcing on LLJ development. Cross-valley winds show a strong signal of nocturnal LLJs starting at about 21 UTC from northern directions and with a maximum intensity between 6 UTC and 9 UTC from NE directions. During the day, a well mixed PBL develops and mean winds turn to S and SW directions. At daytime no LLJ-structure is recognizable in the mean cross-valley wind. The most interesting feature in the mean along-valley wind profiles is the wind component from NW (negative values) developing during the evening transition starting at about 15 UTC with a maximum between 18 UTC and 21 UTC. This flow structure is induced by downslope winds, which are generated after sunset over the steep slopes of the Estrela mountains NW of Perdigão (see Fig. 1(b)). In the course of the night the flow over Perdigão turns from a NW to a N and a NE-flow due to the developing down-valley LLJ along the Castelo Branco basin (see Fig. 10(a)). During the morning transition a SE-flow develops into a S and SW-flow during the day due to upslope and upvalley winds along the Estrela mountain slopes and along the Castelo Branco basin, respectively. This results in a clockwise diurnal wind turning near the surface.

To prove our hypothesis that the diurnal flow structures over Perdigão were thermally driven and not an inertial oscillation phenomenon according to the theory of Blackadar (1957), we computed horizontal gradients of potential temperature $\Theta$ and pressure in cross- and along-valley direction. Three circular regions were defined in Fig. 10 with a diameter of 20 km that were centred at the wind turbine (region A), on the plateau between the Serra da Estrela and Sierra de Gata mountains (region B) and on the steep slopes of the Serra da Estrela mountain range NW of Perdigão (region C). For each region, time series of mean vertical potential temperature and pressure profiles were calculated, which were then used to compute horizontal gradients between region A-B and region A-C. Mean diurnal vertical profiles of these gradients are shown in Fig. 14 averaged over the whole campaign period. The difference of $\Theta$ between region A and B shows higher temperatures between about 9 to 19 UTC and lower temperatures during the night over the elevated plateau in region B (Fig. 14(a)). This can be explained by the valley volume effect (e.g., Wagner, 1938; Whiteman, 2000) and intensified heating/cooling of the PBL over mountain slopes compared to the background atmosphere (Whiteman, 2000). This topographically induced differential warming/cooling leads to a horizontal pressure gradient, which is shown in Fig. 14(c) for the regions A-B and forces a thermally driven SW-flow (higher pressure at region A) during the day and a NE-flow (higher pressure at region B) during the night. A similar, but phase-shifted thermally driven flow system can be observed in along-valley direction when comparing the regions A-C

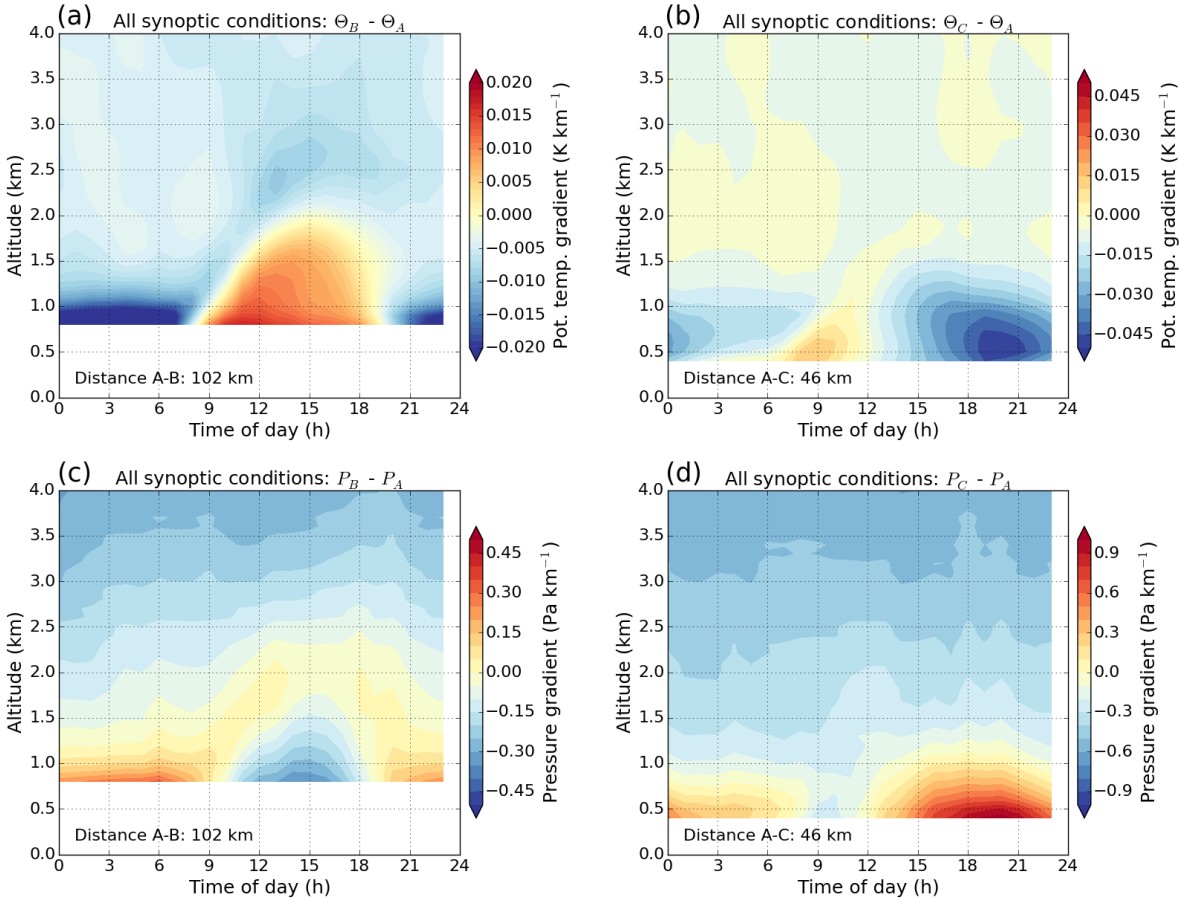

**Figure 14.** WRF D2 mean diurnal vertical profiles of horizontal potential temperature $\Theta$ and pressure $P$ gradients between (a), (c) regions A and B and (b), (d) regions (A) and (C). The location of the regions is marked in Fig. 10. The averaging period was 1 May to 15 June 2017.

in Fig. 14(b) and (d). At late afternoon, the PBL over the steep slopes in region C is cooled and downslope winds from NW directions develop. The positive pressure gradient remains throughout the night and changes sign when the PBL is heated over the mountain slopes in region C after sunrise (Fig. 14(d)). The shown diurnal variation of horizontal pressure gradients is in contrast to the inertial oscillation LLJ-theory of Blackadar (1957). A crucial component of this theory is the existence of a non-vanishing synoptic pressure gradient (e.g., Shapiro and Fedorovich, 2009). During the evening transition a stable boundary layer develops and a LLJ forms due to the sudden reduction of surface friction (Blackadar, 1957; Emeis, 2013). The new balance of pressure-gradient and Coriolis-force induces a clockwise inertial osciallation of the ageostrophic wind component in time. In Fig. 15 a hodograph of the ageostrophic wind components is plotted for three typical LLJ-cases at T20, which occured on 2 May, 3 May and 8 May 2017 over Perdigão. The hodograph does not show the circular oscillation around the zero-point that is typical for the theory of Blackadar (1957). Nearly all points indicate negative ageostrophic wind

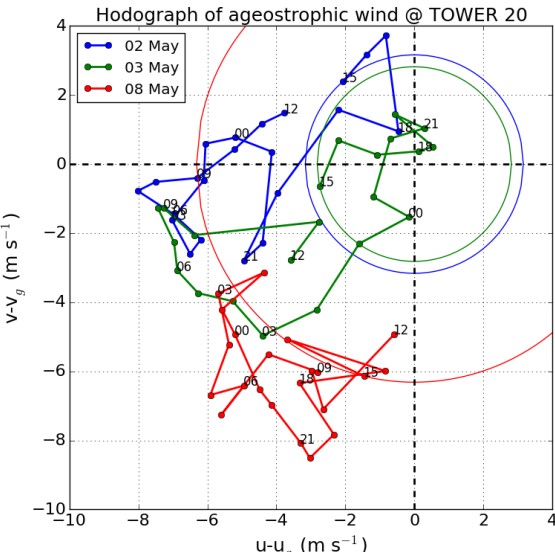

**Figure 15.** Hodograph of ageostrophic wind components for three different nocturnal LLJ-cases (2 May, 3 May and 8 May 2017) at tower T20. Ageostrophic wind components correspond to differences between winds at level $L_1$ and level $L_2$. Numbers indicate the time of the day in hours. Hodographs start at 15 UTC the day before the LLJ-case, respectively (e.g. at 15 UTC on 1 May for the 2 May case). The theoretical inertial osciallations of the ageostrophic wind according to Blackadar (1957) are illustrated by circles for the respective winds at 15 UTC.

components, which means that the wind direction of the LLJs is opposite to the geostrophic wind most of the time. This is confirmed by wind roses of LLJs and winds in the free atmosphere in Fig. 11. It is also in agreement with Fig. 14(c) and (d), which shows a positive nocturnal pressure gradient near the surface and a relatively constant weak negative synoptic pressure gradient above 2.5 km altitude in the free atmosphere. Another hint that the inertial oscillation theory cannot be the main driving mechanism for Perdigão concerns LLJ peak winds that exceed the geostrophic wind by more than 100% (Shapiro and Fedorovich, 2009). This is not possible for inertial oscillations, but was detected in 26% of WRF D3 LLJ cases at T20. This analysis and the good agreement between temperature, pressure and wind patterns in Fig. 13 and in Fig. 14 confirms the dominance of thermally driven flows during the field campaign.

## 6 Conclusions

Long-term WRF-LES simulations were performed with a horizontal resolution of 200 m for a period of 49 days during the Perdigão campaign in May and June 2017. Simulation results were used to characterize the meteorological conditions and to analyse characteristic flow patterns during the intensive observation period (IOP). The high grid resolution of 200 m was necessary to resolve the double-ridge topography. The realistic computation of turbulence features was not the focus of this paper. Large parts of the campaign were dominated by synoptically calm conditions and the evolution of thermally driven flow systems (especially during the second half of the campaign). On the basis of the frequent observation of low-level jets (LLJ) by lidar and in-situ measurements a LLJ-index was computed following the method of Rife et al. (2010) to show that nocturnal LLJs from NE predominantly developed over the steep slopes between the Portuguese Serra da Estrela and the Spanish Sierra de Gata mountain ranges. This katabatic flow intensified during the night and moved down the broad basin of Castelo Branco towards the double-ridge site of Perdigão. During the day, SW winds dominated. Due to the well mixed PBL, this flow had no LLJ-character in most cases. The computation of mean daily cycles of wind direction and cross-valley wind at T20 showed a diurnal clockwise wind turning near the surface with nocturnal LLJs from NE, S- and SW-winds during the day and NW- and N-flow in the evening transition due to downslope winds from the northern Estrela mountain range. This wind turning was also measured by in-situ and lidar instruments. The computation of potential temperature and pressure gradients in cross- and along-valley direction confirmed the hypothesis that PBL-flows were mainly thermally driven during the campaign period. LLJ-generation according to the inertial oscillation theory of Blackadar (1957) did not play a major role due to weak synoptic forcing, opposite pressure gradients in the PBL and in the free atmosphere and missing circular temporal evolution of ageostrophic winds in the LLJ-layer.

The verification of the model with in-situ observations showed a surprisingly good agreement in spite of the long simulation horizon of 49 days. Especially the diurnally changing flow from NE during the night and from SW during the day was captured well on the ridges of the double-hill by **all model domains. This indicates that the mesoscale mountain wind system is generally resolved by a model run with 5 km horizontal grid size. The realistic simulation of the interaction of the flow with the complex topography requires, however, a higher resolution of at least 200 m.** In the valley, observations showed that along-valley winds were the dominating flow regime. WRF D3 simulations computed these along-valley winds, but underestimated the strength of this flow considerably. The reason for this underestimation is not clear. The introduction of turbulence, e.g., by cell perturbation, as well as better landuse data sets including improved roughness and forest maps and

enhanced soil moisture data will likely help to simulate the small scales of valley flows in Perdigão more realistically. This has to be tested in further sensitivity runs. The usage of additional roughness elements, such as the applied forest parameterization, will definitely be necessary for future real-case LES simulations.

*Acknowledgements.* This work was performed within the project LIPS, funded by the Federal Ministry of Economy and Energy on the basis of a resolution of the German Bundestag under the contract numbers 0325518.

We thank José Palma, University of Porto, José Carlos Matos and the INEGI team, as well as the research groups from DTU and NCAR for the successful collaboration and realization of the Perdigão campaign. Additionally, we thank the municipalities of Alvaiade and Vila Velha de Rodão in Portugal for local support. Radiosonde, tower and wind profiler data were kindly provided by NCAR. We appreciate

constructive comments to the manuscript by R. Eichinger and two anonymous reviewers.

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
