# Peer review of "Manuscript prepared for Atmos. Chem. Phys."

_Atmospheric Chemistry and Physics, 2018_

## Referee Comment (RC1) · Anonymous Referee #1 · 9 Oct 2018

General Comments

Using a 49 days long WRL-LES simulation and experimental measurements, the work studies the flow during the intensive observation period of the Perdigão 2017 field campaign. The authors state that during most of the time the flow was thermally driven and used that to study the occurrence of low-level jets. The content is appropriate for ACP, is innovative and the conclusions relevant. It is well described and the results support the conclusions.

Specific Comments

My major concern has to do with coupling WRF with large-eddy simulation in the small-

est domain. I suppose no mechanism was used to generate fine-scale turbulence at the interface, which leads to small-scales having to be generated inside the domain. The dimensions of the domain and the topography may be sufficient for that, but I would like the authors to show some results supporting that the flow over the double-ridge developed realistic turbulence, such as the comparison of results related to the turbulent field in one of the towers (most likely, T20 or T29).

Also, I think that it would be more convenient that Section 4, model verification, was placed before Section 3, where the results of the model are presented and discussed.

Finally, a minor suggestion, is that the authors calculate the correlation between the WRF results and the measurements shown in Figs. 8 and 9 and use that to quantify the quality of the agreement, in the text around line 208.

Technical Corrections

Line 249: Serra da Estrela is incorrectly written "Estrala"

---

## Referee Comment (RC2) · Anonymous Referee #2 · 15 Oct 2018

This is a well written paper describing a very high resolution simulation of the Perdigao field experiment. It certainly required a strong computational investment. It is probably relevant for an ACP special issue on the experiment (I say probably because I have not read the other submitted papers), but I feel it does not contain "a substantial contribution to scientific progress within the scope of this journal", as required by ACP.

Indeed, while the paper mentions a number of relevant mesoscale processes that could be analyzed in detail by 200m resolution simulations (internal waves, low level jets, catabatic flows, to mention a few), there is no detailed analysis of any of those processes, which could be of general interest for atmospheric research. The analysis is

restricted to mean diurnal cycles in specific cross sections, which do show interesting and qualitatively nice responses to the topography, and to the coarse time evolution of a few meteorological variables, which are ok but could be also ok at coarser resolutions. The main conclusion is that the WRF model (with ECMWF high-resolution analysed boundary conditions) performs "well", although there is no indication of its error statistics, or a comparison against a benchmark. The descriptions of the low level flow are rather vague and not supported by specific analysis that could be of general interest.

I understand that such a general paper can make sense in this ACP number, but as an individual paper it makes little sense for ACP as it is. At least I would like to see: (a) a comparison between 200m, 1km simulations and the ECMWF forcing (at 1h): are the higher resolution ones worth the much higher cost? This is a bit technical but of general interest and should be straightforward to do before the paper is accepted; (b) a more through analysis of the low level jet. What process leads to it. Is it an inertial oscillation? What is its typical peak time? How often does it occur and why? I feel this is also important and feasible. (c) some diagnostic of katabatic flows: what is their intensity, location, structure. This mentioned but not really analysed. (d) some diagnostic of internal waves. This may be more difficult, and could be left for future work, but it deserves work.

---

## Author Comment (AC1) · 16 Nov 2018

The comment was uploaded in the form of a supplement:
https://www.atmos-chem-phys-discuss.net/acp-2018-997/acp-2018-997-AC1-supplement.pdf

---

## Author Comment (AC2) · 16 Nov 2018

The comment was uploaded in the form of a supplement:
https://www.atmos-chem-phys-discuss.net/acp-2018-997/acp-2018-997-AC2-supplement.pdf

---

## Author Response (AR1)

**Long-term simulation of the boundary layer flow over the double-ridge site during the Perdigão 2017 field campaign**

**Reply to comments of anonymous referee #1 of manuscript acp-2018-997**

Johannes Wagner et al.

November 16, 2018

**1 Introduction**

We thank the anonymous referee for the comments and acknowledge his effort to improve our manuscript.

In the following, comments of the referee are marked with numbers and corresponding replies of the authors are written in bold and labeled with " $\Rightarrow$ ". In the manuscript we implemented a few new figures (e.g. Fig.4, Fig.7, Fig. 11, Fig. 15) to demonstrate differences in the model results due to different model resolutions (D1 to D3) and to confirm that the main driving mechanism for LLJs were thermally induced pressure gradients and not the inertial oscillation theory of Blackadar (1957). Changes in the new manuscript are written in bold.

**2 Summary**

Using a 49 days long WRF-LES simulation and experimental measurements, the work studies the flow during the intensive observation period of the Perdigão 2017 field campaign. The authors state that during most of the time the flow was thermally driven and used that to study the occurrence of low-level jets. The content is appropriate for ACP, is innovative and the conclusions relevant. It is well described and the results support the conclusions.

**3 Comments**

- 1. My major concern has to do with coupling WRF with large-eddy simulation in the smallest domain. I suppose no mechanism was used to generate fine-scale turbulence at the interface, which leads to small-scales having to be generated inside the domain. The dimensions of the domain and the topography may be sufficient for that, but I would like the authors to show some results supporting that the flow over the double-ridge developed realistic turbulence, such as the comparison of results related to the turbulent field in one of the towers (most likely, T20 or T29).
- $\Rightarrow$  Thank you very much for this comment. Yes, it's right that we did not use a mechanism to generate turbulent perturbations at the lateral edges of our LES-domain such as e.g. the cell-perturbation scheme described in Muñoz-Esparza et al. (2017). We agree that the application of such a method would probably reduce wind speeds of LLJs and could generally improve the model results. The method was, however, not implemented in our model set-up. The intention of our WRF run was not to compute realistic turbulent structures, which are comparable to tower measurements. Our focus was on the simulation of frequently observed daily flow patterns over the double-ridge and to describe their origin and the general meteorological situation during the campaign. In order to resolve the topography properly, a relatively high horizontal model grid resolution of 200 m was necessary. We know that this grid resolution is quite coarse and within the "grey zone" for a LES set-up. Nevertheless, we decided to run domain D3 in LES mode to be independent of a boundary layer parameterization. We used 10 minute values of tower data and set the temporal output interval of our WRF simulations to 10 minutes, respectively. We included this info in section 2 (L52-L57) and emphasized that our focus was not on turbulence characteristics in the conclusions. To illustrate that it is not possible to show realistic turbulence by means of 10 minute tower observations and 10 minute WRF LES data, a power spectrum for crossvalley wind at the location of tower T20 is shown in Fig. 1 (see below). The time resolution of 10 minutes is too coarse to resolve eddies within the inertial subrange. For lower frequencies observed and simulated power curves agree very well.
- 2. Also, I think that it would be more convenient that Section 4, model verification, was placed before Section 3, where the results of the model are presented and discussed.
- $\Rightarrow$  We think that this is a good suggestion and rearranged the sections. After the introduction (section 1) and the model description (section 2) we give an overview of the meteorological situation (section 3). It is followed by the model verification (section 4) and a new section 5 about the LLJ-analysis is placed before the conclusions.

Figure 1: Spectra of cross-valley winds at 100 m AGL for observed and WRF D3 timeseries at tower T20.

- 3. Finally, a minor suggestion, is that the authors calculate the correlation between the WRF results and the measurements shown in Figs. 8 and 9 and use that to quantify the quality of the agreement, in the text around line 208.
- ⇒ We computed correlation coefficients and RMSE-values for the three 100 m towers (T20, T25, T29) for all WRF domains D1 to D3. The values are shown in a new table 2 and the results are described in section 4 (L141-L151).

**4 Technical corrections**

- 4. Line 249: Serra da Estrela is incorrectly written "Estrala"
- $\Rightarrow$  We corrected "Estrela" to "Estrela" in the text.

**References**

Blackadar, A. K.: Boundary Layer Wind Maxima and Their Significance for the Growth of Nocturnal Inversions, Bull. Amer. Meteor. Soc., 38, 283–290, 1957. Muñoz-Esparza, D., Lundquist, J. K., Sauer, J. A., Kosović, B., and Linn, R. R.: Coupled mesoscale-LES modeling of a diurnal cycle during the CWEX-13 field campaign: From weather to boundary-layer eddies, Journal of Advances in Modeling Earth Systems, 9, 1572– 1594, doi:10.1002/2017MS000960, 2017.

**Long-term simulation of the boundary layer flow over the double-ridge site during the Perdigão 2017 field campaign**

**Reply to comments of anonymous referee #2 of manuscript acp-2018-997**

Johannes Wagner et al.

November 16, 2018

**1 Introduction**

We thank the anonymous referee for the comments and acknowledge his effort to improve our manuscript.

In the following, comments of the referee are marked with numbers and corresponding replies of the authors are written in bold and labeled with " $\Rightarrow$ ". In the manuscript we implemented a few new figures (e.g. Fig.4, Fig.7, Fig. 11, Fig. 15) to demonstrate differences in the model results due to different model resolutions (D1 to D3) and to confirm that the main driving mechanism for LLJs were thermally induced pressure gradients and not the inertial oscillation theory of Blackadar (1957). Changes in the new manuscript are written in bold.

**2 Comments**

This is a well written paper describing a very high resolution simulation of the Perdigão field experiment. It certainly required a strong computational investment. It is probably relevant for an ACP special issue on the experiment (I say probably because I have not read the other submitted papers), but I feel it does not contain "a substantial contribution to scientific progress within the scope of this journal", as required by ACP.

Indeed, while the paper mentions a number of relevant mesoscale processes that could be analyzed in detail by 200 m resolution simulations (internal waves, low level jets, catabatic flows, to mention a few), there is no detailed analysis of any of those processes, which could be of general interest for atmospheric research. The analysis is restricted to mean diurnal cycles in specific cross sections, which do show interesting and qualitatively nice responses to the topography, and to the coarse time evolution of a few meteorological variables, which are ok but could be also ok at coarser resolutions. The main conclusion is that the WRF model (with ECMWF high-resolution analysed boundary conditions) performs "well", although there is no indication of its error statistics, or a comparison against a benchmark. The descriptions of the low level flow are rather vague and not supported by specific analysis that could be of general interest.

I understand that such a general paper can make sense in this ACP number, but as an individual paper it makes little sense for ACP as it is.

 $\Rightarrow$  We appreciate your critical comments. The paper was submitted within the ACP special issue "Flow in complex terrain: the Perdigão campaigns" to provide a uniform and continuous data set of meteorological fields throughout the campaign period. This is very useful to understand and interpret flow structures observed in measurement data. It was not the focus of this study to provide realistic turbulence characteristics, but to describe the general meteorological situation and to analyse frequently observed flow phenomena like LLJs. The precondition of such a simulation was to resolve the double-ridge topography, which required a horizontal grid resolution of at least 200 m. To our knowledge a simulation with such a high grid resolution that is conducted over 49 days is not common and worth to be presented to the scientific community. We agree that we did not show comparisons between the 200 m run and results of coarser model domains. We implemented additional figures, described the differences in the text and added a table for correlation coefficients and RMSE-values. We also tried to improve the LLJ-analysis and attempted to demonstrate that the main driving mechanism were thermally induced pressure gradients and not the inertial oscillation theory of Blackadar (1957).

At least I would like to see:

- 1. (a) a comparison between 200m, 1km simulations and the ECMWF forcing (at 1h): are the higher resolution ones worth the much higher cost? This is a bit technical but of general interest and should be straightforward to do before the paper is accepted;
- ⇒ We included comparisons of wind fields for all three WRF domains D1 to D3 (Figs. 4, 7 and 11). We did not include a comparison with ECMWF, as we used the data with a 6 h interval, which would result in wrong interpretations. We showed in the additional table 2 that correlation coefficients cannot be improved by increased model resolutions and that even the coarse domain D1 reproduces the phase of the cross- and along-valley wind signal surprisingly well. RMSE-values are improved by increasing the model resolution, as the

topography becomes more realistic. This becomes clear, e.g. in Fig. 4 and Fig. 7. If only mesoscale flows have to be reproduced, then grid resolutions in the order of 1 km are acceptable. If, however, the interaction of the flow with complex topography should be resolved, 200 m and even finer grids are necessary. This is shown, e.g., for T25 within the valley in Fig. 7.

- 2. (b) a more thorough analysis of the low level jet. What process leads to it. Is it an inertial oscillation? What is its typical peak time? How often does it occur and why? I feel this is also important and feasible.
- ⇒ We introduced a new section 5, which deals with the LLJ-analysis. We show that LLJs occured during 30% of all synoptic conditions and during 42% of NE cases during the campaign at T20 (L231-L233). By means of the new Fig. 11 we show that nearly all detected LLJs come from NE directions and are decoupled from very weak synoptic conditions aloft (L217-225). The typical peak time is between 6 and 9 UTC (L247) as can be seen in Fig. 13 (a) and (c). We tried to identify processes, which might indicate that the inertial oscillation theory of Blackadar (1957) plays a role for the LLJs over Perdigão. We show a hodograph of the ageostrophic wind component for three exemplary LLJ-cases in Fig. 15, which is not in agreement with the circular oscillation of Blackadar (1957). Fig. 14 shows that pressure gradients are not constant in time within the LLJ-layer and can be directed opposite to the synoptic pressure gradient. In 26% of the LLJ-cases winds were more than 100% stronger than synoptic winds, which is also in contrast to the Blackadar (1957) theory (L272-L288).
- 3. (c) some diagnostic of katabatic flows: what is their intensity, location, structure. This is mentioned but not really analysed.
- ⇒ The presented LLJ-index is a measure for the strength of LLJs and in our case also for katabatic flows, as we showed that the jets over Perdigão are thermally driven. Fig. 10 gives an overview of the locations of strongest katabatic flows (L200-L209). In addition, we introduced the new Fig. 11, which demonstrates dominant wind directions of LLJs (katabatic flows) at T20.
- 4. (d) some diagnostic of internal waves. This may be more difficult, and could be left for future work, but it deserves work.
- $\Rightarrow$  We think that this is an interesting suggestion, but requires a thorough and in-depth analysis of the physical processes leading to internal waves and is therefore beyond the scope of this study.

**References**

Blackadar, A. K.: Boundary Layer Wind Maxima and Their Significance for the Growth of Nocturnal Inversions, Bull. Amer. Meteor. Soc., 38, 283–290, 1957.

---

## Author Response (AR2)

**Long-term simulation of the boundary layer flow over the double-ridge site during the Perdigão 2017 field campaign**

**Reply to editor comments of manuscript acp-2018-997**

Johannes Wagner et al.

December 5, 2018

**1 Introduction**

We thank the editor for the comments and acknowledge his effort to improve our manuscript.

In the following, comments of the editor are marked with numbers and corresponding replies of the authors are written in bold and labeled with "⇒". Changes in the new manuscript are written in bold.

**2 Comments**

1. For experiment layout refer to perdigao.fe.up.pt (e.g. page 3)

⇒ **We added the link to the manuscript.**

2. Where it is "Estrala" it should be "Estrela" (e.g. page 13, 15, 23)

⇒ **Thanks for the comment. We misunderstood the technical corrections of reviewer 1 and changed "Estrala" to "Estrela" in the manuscript.**

3. Where it is "Caros" it should be "Carlos" (page 24)

⇒ **We are sorry for this typo and corrected it.**

4. The statement (page 4) "A grid size of 200 m was necessary to properly resolve the double-ridge topography of Perdigão." is not right. 200 m are not enough and the authors should be more cautious.

⇒ **With this sentence we want to say that a resolution of 200 m is necessary to resolve the double-ridge topography with at least 7 grid points. With this resolution the valley and its interaction with the boundary layer can be resolved. We agree that of course much finer computational grids are necessary to simulate the flow within the complex terrain. We added this in the text (L52-L54)**

5. The consequences of relying on WRF alone to mimic the small-scale turbulence requires additional justification, other than "Note that no mechanism was implemented in WRF to generate turbulence at the lateral edges of the LES domain, e.g., similar to the method described in Munoz-Esparza et al. (2017)." (page 4) and "The realistic computation of turbulence features was not the focus of this paper (page 23)". Would the authors please expand their justification and elaborate on the consequences of using the approach by Munoz-Esparza, or in line with referee N. 1 "show ... the comparison of results related to the turbulent field in one of the towers (most likely, T20 or T29)."

⇒ **In our simulations we did not use a technique to introduce turbulence at the edges of the LES-domain, as such methods are not available in the WRF code, yet and as the application of turbulence generating schemes requires higher grid resolutions in the order of 10 m to 50 m (Muñoz-Esparza et al. 2017; Muñoz-Esparza and Kosović 2018). This means that it is not possible to compare simulated and observed turbulence characteristics by using grid resolutions of 200 m. This is visible by means of Fig. 1 (see below), which shows spectra of cross-valley winds at 100 m AGL at tower T20 and indicates that the inertial subrange is not represented. We also added some explanations in the text in L57-L60.**

6. See PDF file attached for additional notes

⇒ **We looked at the additional notes in the PDF and corrected the typos.**

⇒ **We changed the cross-sections in Fig. 4 by plotting larger horizontal distances (-30 km to 30 km for D1 and D2; -15 km to 15 km for D3), as it does not make sense to show distances from -3 km to 4 km for D1 and D2 (see old manuscript).**

**References**

Muñoz-Esparza, D. and Kosović, B.: Generation of Inflow Turbulence in Large-Eddy Simulations of Nonneutral Atmospheric Boundary Layers with the Cell Perturbation Method, Mon. Wea. Rev., 146, 1889–1909, doi:10.1175/MWR-D-18-0077.1, 2018.

[Figure]

Figure 1: Spectra of cross-valley winds at 100 m AGL for observed and WRF D3 timeseries at tower T20.

Muñoz-Esparza, D., Lundquist, J. K., Sauer, J. A., Kosović, B., and Linn, R. R.: Coupled mesoscale-LES modeling of a diurnal cycle during the CWEX-13 field campaign: From weather to boundary-layer eddies, Journal of Advances in Modeling Earth Systems, 9, 1572–1594, doi:10.1002/2017MS000960, 2017.